



# Greenhouse gas column observations from a portable spectrometer in Uganda

Neil Humpage[1,2], Hartmut Boesch[1,2,a], William Okello[3], Jia Chen[4], Florian Dietrich[4], Mark F. Lunt[5,b],
Liang Feng[5,6], Paul I. Palmer[5,6], and Frank Hase[7]

[1]School of Physics and Astronomy, University of Leicester, Leicester, UK
[2]National Centre for Earth Observation (NCEO), University of Leicester, Leicester, UK
[3]National Fisheries Resources Research Institute (NaFIRRI), Jinja, Uganda
[4]Environmental Sensing and Modelling, Technical University of Munich (TUM), Munich, Germany
[5]School of GeoSciences, University of Edinburgh, Edinburgh, UK
[6]National Centre for Earth Observation (NCEO), University of Edinburgh, Edinburgh, UK
[7]Institute of Meteorology and Climate Research (IMK-ASF), Karlsruhe Institute of Technology, Karlsruhe, Germany
[a]now at Institute of Environmental Physics (IUP), University of Bremen, Germany
[b]now at Environmental Defense Fund, Perth, Australia

**Correspondence:** Neil Humpage (nh58@le.ac.uk)

**Abstract.** The extensive terrestrial ecosystems of tropical Africa are a significant store of carbon, and play a key but uncertain role in the atmospheric budgets of carbon dioxide and methane. As ground-based observations in the tropics are scarce compared with other parts of the world, recent studies have instead made use of satellite observations assimilated into atmospheric chemistry and transport models to conclude that methane emissions from this geographical region have increased since 2010 as

a result of increased wetland extent, accounting for up to a third of global methane growth, and that the tropical Africa region dominates net carbon emission across the tropics. These studies rely critically on the accuracy of satellite datasets such as those from OCO-2, GOSAT, and Sentinel-5P TROPOMI, along with results from atmospheric transport models, over a geographical region where there are few independent data to test the robustness of published results.

In this paper we present the first ground-based observations of greenhouse gas column concentrations over East Africa,

obtained using a portable Bruker EM27/SUN FTIR spectrometer during a deployment covering the first few months of 2020 in Jinja, Uganda. We operated the instrument near-autonomously by way of an automated weatherproof enclosure, and observed total atmospheric column concentrations of the greenhouse gases carbon dioxide and methane, as well as carbon monoxide, a useful proxy for emissions from incomplete combustion processes in the region. We discuss the performance of the combined enclosure and spectrometer system that we deployed in Jinja to obtain this data, and show comparisons of our ground-based

observations with satellite datasets from OCO-2 and OCO-3 for carbon dioxide, and Sentinel-5P TROPOMI for methane and carbon monoxide, whilst also comparing our results with concentration data from the GEOS-Chem and CAMS atmospheric inversions that provide a means of increasing spatial and temporal coverage where satellite data are not available. For our measurement period, we find statistically significant differences at the $95\%$ confidence level between the EM27/SUN and OCO-2 $X_{CO_2}$ (OCO-2 lower by a mean of $1.20\,\mathrm{ppm}$, standard deviation $1.05\,\mathrm{ppm}$), and between the EM27/SUN and

Sentinel-5P $X_{CO}$ (Sentinel-5P lower by a mean of $3.68\,\mathrm{ppb}$, standard deviation $7.00\,\mathrm{ppb}$), whilst we found that the differ-





ences between the EM27/SUN and OCO-3 $X_{CO_2}$ (OCO-3 lower by a mean of $1.15\,\mathrm{ppm}$, standard deviation $1.61\,\mathrm{ppm}$), and between the EM27/SUN and Sentinel-5P $X_{CH_4}$ (Sentinel-5P lower by a mean of $8.33\,\mathrm{ppb}$, standard deviation $10.5\,\mathrm{ppb}$), were not statistically significant. With regards to the model comparisons, we also see statistically significant differences between the EM27/SUN and a global GEOS-Chem inversion for $X_{CO_2}$ (GEOS-Chem lower by a mean of $0.35\,\mathrm{ppm}$, standard deviation

$1.08\,\mathrm{ppm}$), between the EM27/SUN and a high-resolution GEOS-Chem inversion for $X_{CH_4}$ (GEOS-Chem lower by a mean of $3.80\,\mathrm{ppb}$, standard deviation $12.5\,\mathrm{ppb}$), and between the EM27/SUN and CAMS global analysis $X_{CO}$ (CAMS lower by a mean of $11.7\,\mathrm{ppb}$, standard deviation $8.94\,\mathrm{ppb}$). Our results demonstrate the value of ground-based observations of total column concentrations, and show that the combined EM27/SUN and enclosure system employed would be suitable for acquisition of the longer-term observations needed to rigorously evaluate satellite observations and model calculations over tropical

Africa.

## 1 Introduction

Gaps in our understanding of the global carbon cycle add uncertainty to our predictions of future climate change, including how the future climate will respond to different carbon emissions scenarios (IPCC, 2021; Friedlingstein et al., 2022). One part of the carbon cycle which still requires further investigation is that of carbon fluxes from terrestrial tropical ecosystems, which

store large quantities of carbon in vegetation and soil whilst being sensitive to changes in the climate (Pan et al., 2011; Crowther et al., 2015). Carbon dioxide is released by these ecosystems into the atmosphere through a combination of respiration and fire, and is removed from the atmosphere by photosynthesis and subsequent conversion into plant biomass. The tropics are also home to extensive areas of wetlands, which are the most significant natural source of methane in the atmosphere via the decomposition of organic matter in anaerobic conditions (Kayranli et al., 2010; Mitsch et al., 2013). Further microbial sources

of methane in tropical regions include agricultural practices, particularly the farming of ruminants, and waste disposal. Looking at Africa in particular, an additional factor having an increasingly significant impact on the tropical African carbon cycle is the recent increase in population in many African countries resulting in increasing demand for energy (Ayompe et al., 2021), as reflected in the rapid projected growth of cities such as Kampala (Uganda), Nairobi (Kenya) and Kinshasa (Democratic Republic of Congo). This combination of natural and anthropogenic fluxes that contribute to the atmospheric carbon budget in

tropical Africa is challenging to accurately represent in climate and atmospheric chemistry models, so we need to make use of atmospheric composition measurements to evaluate our understanding.

Compared with other parts of the world, however, ground-based measurements of atmospheric composition are scarce in tropical Africa, placing an upper limit on how well we can understand the carbon cycle in this region (López-Ballesteros et al., 2018; Nickless et al., 2020). This measurement gap is partially addressed by satellites such as the JAXA GOSAT (Kuze

et al., 2009), NASA OCO-2 (Eldering et al., 2017) and OCO-3 (Eldering et al., 2019), and Copernicus Sentinel-5P (Veefkind et al., 2012) missions, although the measurement technique employed usually requires cloud-free and low aerosol conditions to retrieve molecular concentrations from the observed radiance spectra, resulting in relatively poor coverage over the tropics where conditions are often cloudy. The satellite data that are obtained can however be used as an input for atmospheric



chemistry models, which use prior estimates of surface fluxes and meteorological fields to calculate a most-likely state for the
atmosphere constrained by the observations available (e.g. Basu et al. (2013); Deng et al. (2014); Feng et al. (2017); Chevallier
et al. (2019); Crowell et al. (2019); Chen et al. (2022); Peiro et al. (2022)). The atmospheric chemistry models can then be
used in an inversion framework to produce *a posteriori* estimates of emissions where satellite data are not available, since
increased greenhouse gas concentrations remain in the atmospheric column for some time downwind of where they are origi-
nally emitted. In addition, these models (such as GEOS-Chem: Turner et al. (2015); Feng et al. (2017); Lunt et al. (2019))
are a useful means for estimating atmospheric concentrations where observations are not available, and have underpinned a
number of studies that address the tropical African carbon cycle (Palmer et al., 2019; Lunt et al., 2019, 2021; Pandey et al.,
2021; Qu et al., 2022; Feng et al., 2022, 2023; Drinkwater et al., 2023). It is therefore important to validate model output with
independent observations, to confirm how well the models represent the atmosphere and add weight to the conclusions of the
studies which use them.

In this study, we describe observations of the total column concentrations of greenhouse gases in Uganda in the first few
months of 2020, obtained using a portable spectrometer with a built-in solar tracker. We used an automated enclosure to provide
a weatherproof environment for the spectrometer, and to allow us to operate the spectrometer remotely. This setup allowed us
to produce for the first time a dataset of ground-based total column concentrations of carbon dioxide, methane, and carbon
monoxide for a tropical East African location. In Section 2 we outline the measurement site and describe the instrument,
enclosure, and retrieval algorithm used to obtain the dataset. Section 3 covers the satellite and model datasets that we then
compare with our ground-based observations in Section 4. We finally conclude and consider the implications of this study in
Section 5.

## 2  The measurement site at NaFIRRI in Jinja, Uganda

For this study, we established our measurement site at the headquarters of the Ugandan National Fisheries Resources Research
Institute (NaFIRRI) in Jinja ($0.4165°$N, $33.2070°$E, 1157 metres above sea level). Jinja is located on the northern shore of
Lake Victoria, approximately 70 km to the east of Kampala, Uganda's capital city with a population of approximately 3.5
million people across its wider urban area. The source of the White Nile is in Jinja, from which it flows northwards out of Lake
Victoria, through Lakes Kyoga and Albert, and onwards into South Sudan. The Nile feeds the neighbouring wetlands, which
are amongst the main sources of methane emissions in the East Africa through the anaerobic decomposition of organic matter.
There is a strong link between $CH_4$ emissions and water table depth in tropical regions, such that anomalies in precipitation
can lead to wetland $CH_4$ emissions anomalies (Bloom et al., 2010). The hydrological flow from increased precipitation over
Lake Victoria to higher water table depth in Ugandan and South Sudanese wetlands, as a result of increased volumes of water
transported along the Nile, is covered in more detail by Lunt et al. (2019). Precipitation in Uganda is driven by the annual north-
south movement of the inter-tropical convergence zone, resulting in two main wet seasons during the year: these are known
as the 'long rains' which occur from March to May, and the 'short rains' occurring from October to December (Herrmann
and Mohr, 2011). The amount of precipitation over East Africa during these wet seasons is in turn partly influenced by ocean



temperatures in the Indian Ocean (Palmer et al., 2023), where an unusually high contrast in temperatures (greater than $0.4°C$) between the warmer western Indian Ocean and cooler Eastern Indian Ocean (defined quantitatively as the Indian Ocean Dipole, Saji et al. (1999)) in 2019 resulted in one of the wettest short rains seasons on record (Wainwright et al., 2021).

In the wider region beyond Uganda, there are a number of environmental factors which can potentially affect the column of air that we measure at Jinja. To the north, the Sudd wetlands in South Sudan represent a significant natural source of $CH_4$ as discussed and investigated by Lunt et al. (2019) and Lunt et al. (2021). To the west, atmospheric $CO_2$ signals are dominated by the biospheric influence of the Congo rainforest (Palmer et al., 2019). This part of the world is also subject to a high frequency of biomass burning events, evidence of which can be seen in TROPOMI observations of carbon monoxide (a

product of incomplete combustion, see Section 3.2).

To help understand which sources influence the composition of the air columns we observe, we can use a Lagrangian dispersion model to calculate the history of the air masses arriving over Jinja during the measurement period (Fleming et al., 2012; Panagi et al., 2020). We use the UK Met Office's (UKMO) Numerical Atmospheric Modelling Environment (NAME) to perform this task. NAME, along with other Lagrangian dispersion models, works by releasing a large number of inert

particles from a specific location in the atmosphere and then tracking their pathways backwards in time using meteorological model data (which in the NAME modelling framework comes from the UKMO Unified Model). We then count the number of released particles that pass within $100\,\mathrm{m}$ of the surface over each spatial grid point, to determine where and to what extent the back trajectories are influenced by surface emissions from that location within a certain period of time. To account for our measurements being sensitive to the whole atmospheric column, we perform the particle releases from heights throughout the

vertical grid of the model domain, and weight the contributions from each release height according to the pressure weighting function of the EM27/SUN observations. We perform the calculation for each day of the measurement period (performing the particle release at 1030 UTC each day, equivalent to 1330 local time – the time at which Sentinel-5P and OCO-2 pass overhead – and tracking it back in time for one day and five days) to obtain a daily column footprint. Figure 1 shows the mean daily footprint for the whole measurement period (panels A and B showing the results from back trajectories going back one and

five days, respectively), giving us an estimate of where the surface has influenced the measured column. Although the highest contribution arrives via a region directly to the south, coinciding with Lake Victoria, the footprint of influence also covers regions to the north and east, reaching as far as South Sudan and Kenya respectively, where emissions from wetlands and agriculture can potentially have an impact on the observed atmospheric column.

## 2.1 The EM27/SUN portable spectrometer

Ground-based remote sensing of the atmospheric column has proved to be an invaluable tool in the validation of atmospheric composition data from satellite observations. The global network of Bruker 125HR spectrometers that form TCCON (Total Carbon Column Observing Network), for example, is now routinely used in the validation of greenhouse gas column observations from GOSAT, OCO-2, Sentinel-5P and others, allowing those working on the retrieval algorithms to identify, and correct for, systematic biases in their data. The standard configuration for a TCCON site is, however, both expensive and logistically

challenging to set up and maintain. As a result, there are certain regions around the world – South America, Africa, and





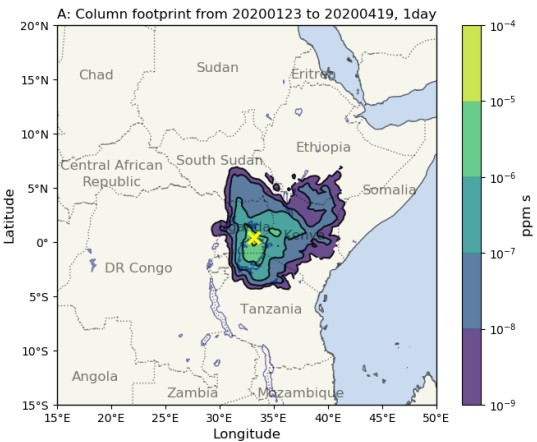
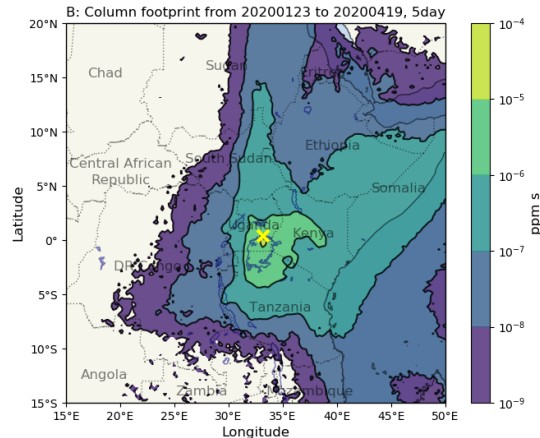

**Figure 1.** Mean column footprint of 1030 UTC (1330 local time) column observations, calculated from 1 day (panel A) and 5 day (panel B) NAME back trajectories for each day within the measurement period (23$^{rd}$ January to 19$^{th}$ April 2020). The yellow cross indicates the location of our EM27/SUN instrument in Jinja. The colour scale indicates the calculated contribution of the surface in that location to the observed atmospheric column, in ppm, integrated over the duration of the back trajectory.

Central/Southern Asia – which do not currently have the resources and infrastructure in place to host TCCON sites, leaving significant gaps in the validation of GHG column data products, often in geographical areas of great scientific interest (e.g. the Amazon rainforest, sub-Saharan Africa).

The Bruker EM27/SUN FTIR (Fourier Transform InfraRed) spectrometer concept (Gisi et al., 2012) was developed at the
Karlsruhe Institute of Technology (KIT), in part to address this problem. It comprises a portable Fourier transform spectrometer with built-in solar tracker, which trades off a reduced spectral resolution compared with the Bruker 125HR used at TCCON sites in favour of being less expensive, and much easier to transport to and operate in different locations. A number of previous studies have demonstrated comparable stability and precision when operated side-by-side with the higher resolution 125HR (Frey et al., 2015; Hedelius et al., 2016; Hase et al., 2016; Sha et al., 2020; Alberti et al., 2022). At the time of writing,
over 150 EM27/SUNs have been purchased by research groups around the world, and operated in a variety of locations. Prior to shipment, the instruments are first calibrated at KIT to obtain the instrument line shape parameters, and are operated side-by-side with a reference instrument in Karlsruhe to derive instrument-specific scaling factors, which can be applied by the user to their retrieved GHG column data to maintain consistency between all EM27/SUN data sets, regardless of who operates the instrument and where. This work is done under the COCCON (COllaborative Carbon Column Observing Network: Frey et al.
(2019); Alberti et al. (2022)) project, which also develops and maintains the PROFFAST retrieval software used to calculate atmospheric column concentrations from the measured interferograms.

As well as being used for validation studies in various locations (Jacobs et al., 2020; Tu et al., 2020; Frey et al., 2021), the portability and relatively low cost of the EM27/SUN has lead to a variety of other scientific applications. In the city of



Munich, Germany, a permanent network of five EM27/SUNs has been established to observe the city's carbon emissions
using the differential column observation method (Chen et al., 2016; Dietrich et al., 2021), and provide a means of validating
spatial gradients in OCO-2 target mode observations of $X_{CO_2}$ (Rißmann et al., 2022). Similar city-focused studies using
EM27/SUNs have taken place in Berlin (Hase et al., 2015), St. Petersburg (Makarova et al., 2021), Beijing (Che et al., 2022;
Zhou et al., 2022), and Indianapolis (Jones et al., 2021) amongst others. Further studies have taken advantage of the instrument's
portability in another way, adapting the instrument with a specially designed solar tracker for operation on board a cargo ship
to provide a unique opportunity for validation of satellite and model data over the ocean (Klappenbach et al., 2015; Knapp
et al., 2021). Some of the studies listed here make use of various designs of weatherproof enclosure to operate the EM27/SUN
more effectively; the enclosure we use here, developed at TU Munich, is described in Section 2.2.

## 2.2    An automated enclosure for the EM27/SUN

The EM27/SUN, whilst very useful for greenhouse gas column observations, is not suitable for unattended operations 'out of
the box'. Firstly the instrument itself is not weatherproof, so the user has to keep a close eye on the weather forecast when
deciding whether to set up for a day of measurements, and be in close attendance to move it indoors in case of rain. In addition,
the user has to manually start the solar tracker and then the spectrometer at the beginning of each day, before powering down
and moving it indoors once the day's observations are complete. This labour intensive mode of operations works well for short
term measurement campaigns, but is less suitable if the goal is to obtain long term observations in a single location.

To make the EM27/SUN suitable for use on longer term deployments, the Environmental Sensing and Modelling Group at
the Technical University of Munich have developed an automated enclosure (Heinle and Chen, 2018; Dietrich et al., 2021)
which provides weatherproofing, environmental control, and automation of the observations. The main components of the TU
Munich enclosure are labelled in Figure 2, and are described in detail by Dietrich et al. (2021).

A modified Zarges K470 aluminium box is used for the main body of the enclosure system. On top of the housing, a rotating
cover closes to protect the contents of the enclosure system when rain is detected by an optical rain sensor, and overnight when
no measurements are taking place. When conditions are dry during the day time, the cover rotates to track the azimuth angle
of the solar tracker and allow sunlight into the system.

The user controls and monitors the enclosure system by remotely accessing the enclosure computer, which also controls
the EM27/SUN spectrometer and solar tracker, and stores the measured interferograms. The automated features of the system
are controlled by a programmable logic controller (PLC), ensuring that critical safety features protecting the system (detection
of rain or power failure, control of the cover motor, temperature control) are not dependent on the enclosure computer. An
additional challenge posed by the location of these measurements very close to the equator is that of very high solar zenith
angles, which at times are beyond the normal operating range of both the solar tracker and the protective cover. A pair of
car-jacks attached to the side of the enclosure (see Figure 2) allow the entire enclosure system to be tilted, such that the sun
can be tracked throughout the middle of the day.

Control and automation of the enclosure system is achieved by two software programs, both developed in-house at TU
Munich (Dietrich et al., 2021). The first of these (Enclosure Control, or ECon: see Heinle and Chen (2018)) controls the





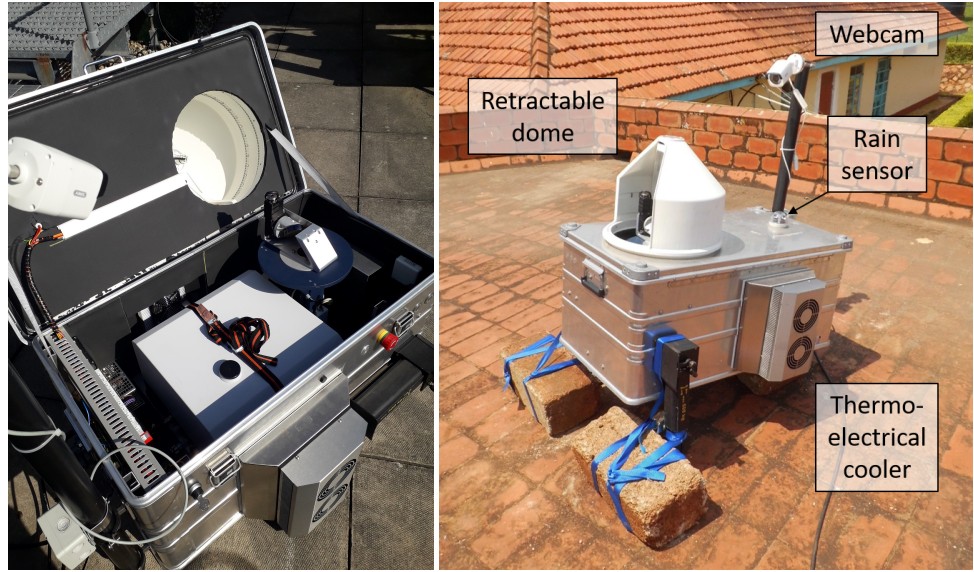

**Figure 2.** Left: internal view of the EM27/SUN spectrometer and solar tracker housed within the TU Munich enclosure system; Right: the enclosure system in operation at NaFIRRI, with the two car-jacks used to tilt the system to enable tracking of the sun at very high solar zenith angles (see text in Section 2.2). The bricks attached to the car-jacks anchor the enclosure down in case of strong winds.

enclosure itself – moving the rotating cover into the correct position, maintaining internal temperature using the thermo-electrical coolers, monitoring the rain sensor data and the UPS, and powering the spectrometer. ECon also checks that the

Ethernet connections linking the different components of the enclosure system are working correctly, and performs automatic restarts or specific components if a malfunction is detected. Alongside ECon, control of the spectrometer and solar tracker are automated using a Python program called Pyra. Pyra effectively acts as a wrapper for the software provided by Bruker that controls the spectrometer and the solar tracker (OPUS and CamTracker, respectively), providing the means to start, stop, and control them automatically. For these measurements we used Pyra in a semi-automated mode which started and stopped the

observations when the solar zenith angle passed a minimum threshold; a more detailed description of Pyra can be found in Appendix A of Dietrich et al. (2021), whilst the latest version is described in full by Aigner et al. (2023).

### 2.3 Total column concentrations over Jinja from the EM27/SUN and automated enclosure system

The data processing method we use, taking us from the raw interferograms measured by the EM27/SUN spectrometer to the column averaged greenhouse gas abundances over Jinja presented in this paper, is described in more detail in Section 2.2 of

Frey et al. (2021). The method comprises two parts, both written in FORTRAN: PREPROCESS, which performs Fast Fourier Transforms on the interferograms (that are first corrected for intensity fluctuations and apodised) to obtain radiance spectra; and PROFFAST, which then retrieves the column averaged greenhouse gas abundances from the radiance spectra. Several quality filters, summarised in Table 1 of Frey et al. (2021), are applied to each interferogram by the PREPROCESS routine.





The *a priori* profiles we use for trace gas concentrations, pressure, and temperature, are those generated for use in the TCCON
GGG2014 data version (Wunch et al., 2015).

We then apply the PROFFAST retrieval algorithm to those spectra whose interferograms have passed the quality filters applied during the PREPROCESS stage. PROFFAST is a non-linear least squares algorithm which scales *a priori* trace gas profiles to fit forward modelled atmospheric spectra to the measured spectra, and then calculates the retrieved total column abundances from the scaled profiles. These are finally converted into column-averaged dry-air mole fractions $X_{\mathrm{gas}}$, given by

$$X_{\mathrm{gas}} = \frac{\mathrm{VC}_{\mathrm{gas}}}{\mathrm{VC}_{\mathrm{O}_2}} \times 0.2095, \tag{1}$$

where $\mathrm{VC}_{\mathrm{gas}}$ is the retrieved total column abundance for that gas. Taking the ratio of the total column abundances has the benefit of at least partially cancelling out any spectroscopic errors which affect both VCs in a similar way (Wunch et al., 2010), whilst also reducing the dependence on the measured ground pressure (Frey et al., 2021).

To monitor the stability of the spectrometer, we use the column-averaged amount of dry air ($X_{\mathrm{air}}$). This is the ratio of the
total column abundance of dry air calculated from the retrieved total column abundance of oxygen, $\mathrm{VC}_{\mathrm{O}_2}$, to the total column abundance of dry air calculated from the measured surface pressure $P_{\mathrm{S}}$, and is given by

$$X_{\mathrm{air}} = \frac{g}{P_{\mathrm{S}}} \cdot \left( \frac{\mathrm{VC}_{\mathrm{O}_2} \cdot \overline{\mu}}{0.2095} + \mathrm{VC}_{\mathrm{H}_2\mathrm{O}} \cdot \mu_{\mathrm{H}_2\mathrm{O}} \right), \tag{2}$$

where the molecular masses of dry air and water vapour are given by $\overline{\mu}$ and $\mu_{\mathrm{H}_2\mathrm{O}}$ respectively, $g$ is the column-averaged gravitational acceleration, and $\mathrm{VC}_{\mathrm{H}_2\mathrm{O}}$ is the retrieved water vapour total column (this correction is required to allow for the
measured surface pressure including the whole air column, whereas it is the dry air column that we retrieve using the oxygen absorption band). As long as the spectrometer is working nominally, $X_{\mathrm{air}}$ should remain close to 1.0 and stable over time. We therefore use $X_{\mathrm{air}}$ as a final quality filter on the retrieved column data, by removing any data points where the difference from the daily median value of $X_{\mathrm{air}}$ is greater than 0.002, and then removing any further data points which deviate from the rolling hourly mean $X_{\mathrm{air}}$ by more than 0.0005.

A final step in the data processing is to apply calibration factors to the retrieved column concentrations, which bring the results into line with the rest of the EM27/SUNs involved in COCCON (as discussed earlier in this Section, a full list of calibration factors is given in Table 6 of Frey et al. (2019)). For the spectrometer used here (serial number 059), the calibration factors with respect to the reference EM27/SUN operated by Karlsruhe Institute of Technology (serial number 037) are 0.9998, 0.9991 and 1.0019 for $X_{\mathrm{CO}_2}$, $X_{\mathrm{CH}_4}$ and $\mathrm{O}_2$, respectively. The column concentrations retrieved by following the procedure
described above are shown in Figure 3, along with the number of quality-controlled soundings obtained on each measurement date. The daily count of measurements leading to a valid retrieval is determined by a combination of weather conditions (cloudy vs. cloud-free) and the availability of mains power on the NaFIRRI site, during the times of day when the Sun is at least 20° above the horizon.

**Figure 3.** From top to bottom: column concentrations of $CO_2$, $CH_4$, $CO$, and $H_2O$ retrieved from the EM27/SUN measurements using the PROFFAST algorithm as described in Section 2.3; surface air temperature; surface pressure; and relative humidity. The solid line in the upper four panels indicates the number of successful retrievals on each day, for each gas. The vertical dashed line marks the onset of the 'long rains' as described in Section 2.





## 3 Satellite and model datasets used in this study

In this section, we introduce the satellite and model datasets that we compare with our Jinja EM27/SUN column concentration data.

### 3.1 Orbiting Carbon Observatory (OCO-2 and OCO-3)

The Orbiting Carbon Observatory-2 (OCO-2) was launched in 2014, and was specifically designed by NASA to have the precision required to detect the changes in $X_{CO_2}$ that correspond to surface emissions and uptake of $CO_2$, on a regional scale with

global coverage (Eldering et al., 2017). The sole OCO-2 payload comprises a three-band grating spectrometer, which measures the radiance spectra of sunlight reflected back into space by the Earth's surface. Of the three spectral bands, two coincide with carbon dioxide absorption features (the so-called 'weak' and 'strong' $CO_2$ bands, centred at wavelengths of 1.6 and 2.0 $\mu$m respectively), whilst the third band at 0.76 $\mu$m is used to measure absorption by molecular oxygen. The instrument samples eight spatial footprints across-track, which are each nominally 1.25 km in width at the surface. Along-track, each footprint is

around 2.4 km in length owing to the distance travelled by the satellite during the instrument's 0.33 s integration time. The orbit track and the narrow swath width (approximately 10 km wide) mean that the same ground location is resampled once every 16 days. A full-physics retrieval algorithm based on an optimal estimation technique is used to retrieve $X_{CO_2}$ from the OCO-2 measured spectra (O'Dell et al., 2012, 2018), taking into account multiple scattering and polarisation effects. The retrieved column concentrations are validated against the TCCON ground-based network of Bruker 125HR spectrometers (Wunch et al.,

2017). For this study, we use Version 10r of the OCO-2 data (Taylor et al., 2023) – the spatially gridded mean $CO_2$ column concentrations from this dataset observed over East Africa during our measurement period are shown in Figure 4A.

In 2019 NASA also integrated the flight spare for OCO-2, under the name Orbiting Carbon Observatory-3 (OCO-3, Eldering et al. (2019); Taylor et al. (2020)), onto the International Space Station (ISS). The low-inclination orbit occupied by the ISS introduces significant differences to the sampling pattern, with the main implication being that overpasses of a particular

location do not take place at the same local time each day. The overpass time instead shifts about 20 minutes earlier from one day to the next, such that all times of day are eventually observed. In contrast to the observations obtained from the sun-synchronous orbit followed by OCO-2, this means that OCO-3 can provide information on how $X_{CO_2}$ varies with time of day. In addition, the Pointing Mirror Assembly (PMA) allows pointing towards the ocean glint spot to maximise the observed signal over water (for OCO-2 the whole spacecraft is rotated to achieve a similar goal), or towards stationary ground targets

such as validation sites. Uniquely to OCO-3, snapshot area maps (SAMs) can also be acquired. These involve sweeping the PMA fore-optics back and forth across an area approximately 85 km × 85 km in size, effectively producing spatially resolved 2D images of $X_{CO_2}$ over areas of interest. The same full-physics retrieval algorithm is used on OCO-3 measurements as for OCO-2 to obtain the column concentrations of $CO_2$. Here we use Version 10.4r of the OCO-3 data (Taylor et al., 2023). The gridded mean $X_{CO_2}$ over East Africa for our whole measurement period is shown in Figure 4B, which clearly illustrates the

different spatial sampling pattern employed by OCO-3 compared with OCO-2 (Figure 4A). Note that Jinja is included on the




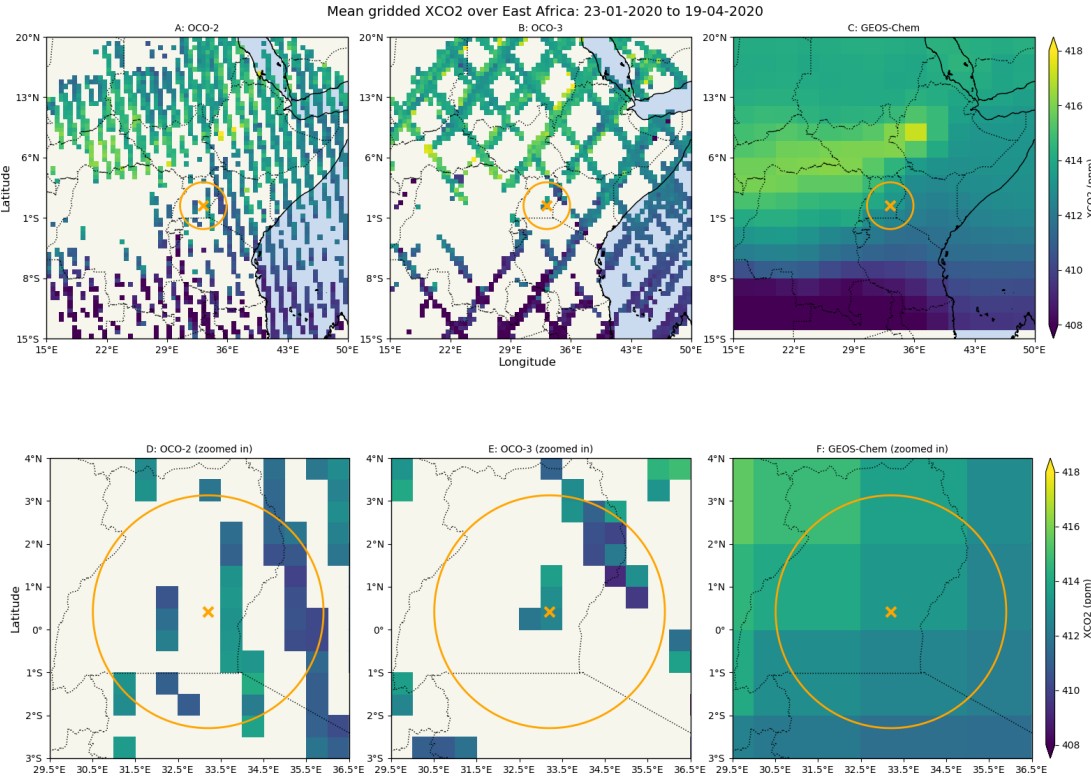

**Figure 4.** $X_{CO_2}$ over East Africa retrieved from OCO-2 (panel A) and OCO-3 (panel B) observations, averaged over the EM27/SUN measurement period (23$^{rd}$ January to 19$^{th}$ April 2020) and spatially binned into a 1x1 degree grid. The orange cross shows the location of the measurement site in Jinja, and the circle indicates the co-location criteria (300 km radius) used in the comparisons described in Section 4.1. Panel C shows the $X_{CO_2}$ output from the global GEOS-Chem inversion described in Section 3.3. Panels D, E and F show the same data as panels A, B and C respectively, zoomed in on the co-location region.

list of targets for the SAM mode, such that the OCO-3 soundings taken as the ISS passes over Uganda tend to be concentrated within a short distance of our measurement site.

## 3.2   Sentinel-5 Precursor TROPOMI

The Copernicus Sentinel-5 Precursor (S5P) mission was launched in October 2017 to measure atmospheric composition,
specifically air quality and climate change indicators, with daily global coverage and moderately high (up to $5.5 \text{ km} \times 3.5 \text{ km}$ at nadir) spatial resolution. The sole payload of the S5P mission is the TROPOspheric Monitoring Instrument (TROPOMI, Veefkind et al. (2012)), a grating spectrometer with four spectral bands covering ultraviolet, visible, near-infrared (NIR), and shortwave infrared (SWIR) wavelength ranges, respectively. The S5P operational $CH_4$ retrieval algorithm uses the NIR and





SWIR spectral channels in a full-physics, optimal estimation method to derive the column-averaged dry mixing ratio of methane ($X_{CH_4}$) from the TROPOMI measurements (Hu et al., 2016; Hasekamp et al., 2021). The $X_{CH_4}$ data used in this study has been processed using version 01.03.02 of the TROPOMI $CH_4$ processor, which has been shown to perform well within the mission requirements through comparison with ground-based observations from 28 TCCON stations (Sha et al., 2021). This version includes a surface albedo dependent *a posteriori* bias correction, based on comparisons between co-located S5P and GOSAT $X_{CH_4}$ data (Hasekamp et al., 2021). We show the gridded mean $X_{CH_4}$ from this dataset over East Africa, averaging over our whole measurement period, in Figure 5A. More recent TROPOMI $X_{CH_4}$ data (measurements from 1st July 2021 onwards) use version 2 of the processor, which incorporates a number of improvements including updated $CH_4$, CO and $H_2O$ spectroscopic cross sections, and an updated *a posteriori* bias correction that is independent of external reference data (Lorente et al., 2021).

The S5P TROPOMI SWIR band is also used to retrieve the total column abundances of carbon monoxide, $X_{CO}$ (Landgraf et al., 2016). The retrieval uses a two-step process: firstly, a non-scattering retrieval of the total amount of $CH_4$ is performed, and then compared with modelled methane abundances to act as a cloud filter (if the retrieved $CH_4$ assuming no-scattering differs significantly from the model value, this indicates that the impact of scattering from high or optically thick cloud is too great to perform a useful carbon monoxide retrieval). The second step then retrieves the CO column using a profile scaling approach, along with the $H_2O$ abundance and effective cloud parameters using the *a priori* knowledge of methane acquired during the first step. Validation of the operational S5P TROPOMI $X_{CO}$ against ground-based TCCON observations has demonstrated that the requirements for systematic and random uncertainties in the data are being met (Sha et al., 2021). Figure 6A shows the gridded mean $X_{CO}$ from the operational TROPOMI product for our measurement period, over the East Africa region. The less strict requirements on accuracy and precision for $X_{CO}$ compared with those for $X_{CH_4}$ allows retrievals to be made over land and ocean scenes, under both clear-sky and (with the exception of high or optically thick clouds) cloudy conditions. This is reflected in the comprehensive spatial coverage shown in Figure 6A that is achieved by the $X_{CO}$ retrieval compared with that of $X_{CH_4}$ (as seen in Figure 5A), which requires cloud-free conditions and minimal scattering for a successful retrieval.

### 3.3 GEOS-Chem and CAMS concentration data

GEOS-Chem is an atmospheric chemistry transport model that is used here to simulate the emissions, sinks, chemistry, and transport of carbon dioxide and methane (Turner et al., 2015; Feng et al., 2017; Lunt et al., 2019, 2021), and produce three-dimensional fields of their concentrations. This can provide a useful extension of satellite data in spatial regions and at times of day where the satellite data aren't available. For a more detailed description of the GEOS-Chem model and the ensemble Kalman filter inverse method used, we refer the reader to the papers cited below.

For carbon dioxide, we use a global GEOS-Chem model run on a $2.0° \times 2.5°$ latitude-longitude grid with 47 vertical levels. We use emissions inventories for our *a priori* flux estimates, taking into account $CO_2$ emissions from biomass burning (van der Werf et al., 2010), fossil fuels (Oda et al., 2018), ocean fluxes (Takahashi et al., 2009) and biosphere fluxes (Olsen and Randerson, 2004). An ensemble Kalman Filter approach is then used to estimate the $CO_2$ fluxes, with either *in-situ* or satellite measurements of atmospheric $CO_2$ used as prior information on concentration (Feng et al., 2009; Palmer et al., 2019). The mean $X_{CO_2}$ values for the measurement period calculated from the output of this global inversion are shown in Figure 4C.

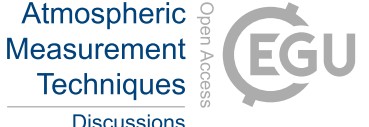

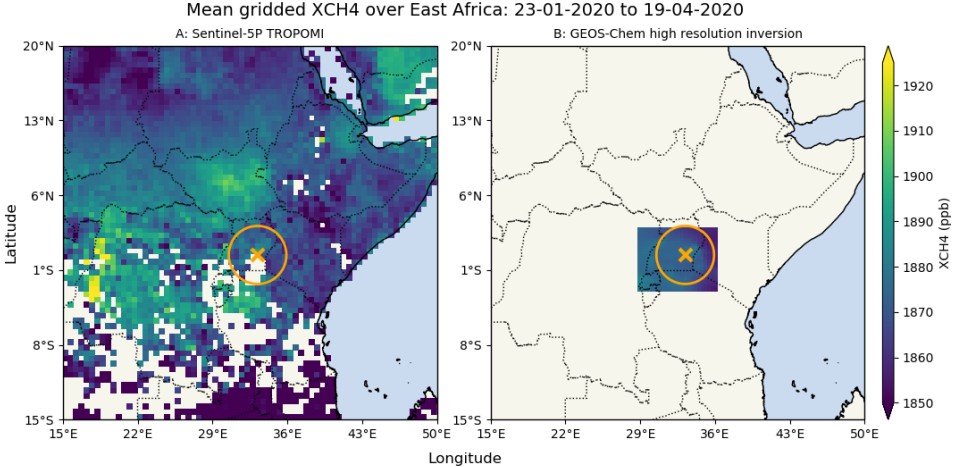

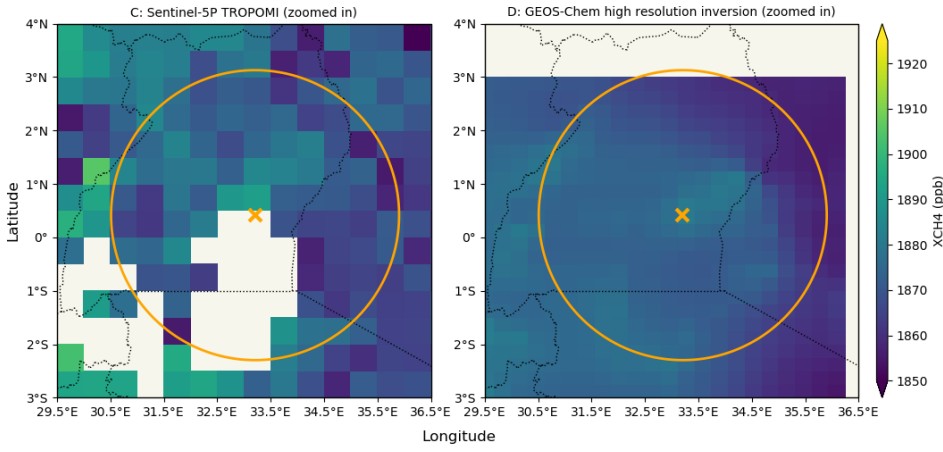

**Figure 5.** $X_{CH_4}$ over East Africa retrieved from Sentinel-5P TROPOMI observations (Panel A, see Section 3.2) and calculated from a high resolution GEOS-Chem inversion (Panel B, see Section 3.3 for details), averaged over the EM27/SUN measurement period (23[rd] January to 19[th] April 2020). The TROPOMI data are spatially binned into a 1x1 degree grid. The orange cross shows the location of the measurement site in Jinja, and the circles indicate the co-location criteria (300 km radius for $X_{CH_4}$) used in the comparisons described in Section 4.2. Panels C and D show the same data as panels A and B respectively, zoomed in on the co-location region.

In the case of methane, we run GEOS-Chem in a nested configuration at high spatial resolution ($0.25° \times 0.3125°$) over a latitude-longitude box covering sub-Saharan Africa ($-36.0$ to $+20.0°$N, $-20.0$ to $55.0°$E), using the setup described in detail by Lunt et al. (2021). The inversion analysis we show here is an extension of the inversion presented in Lunt et al. (2021), from the end of 2019 through the first four months of 2020. For the *a priori* methane emissions inside the nested domain we use the





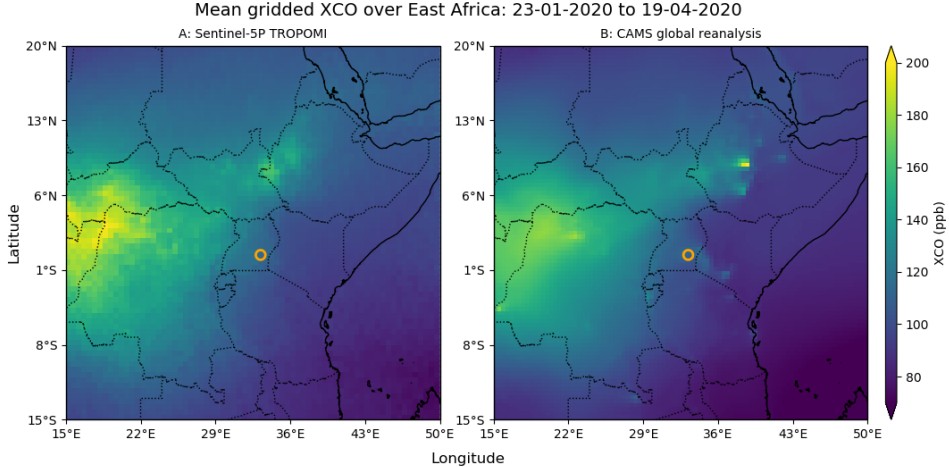

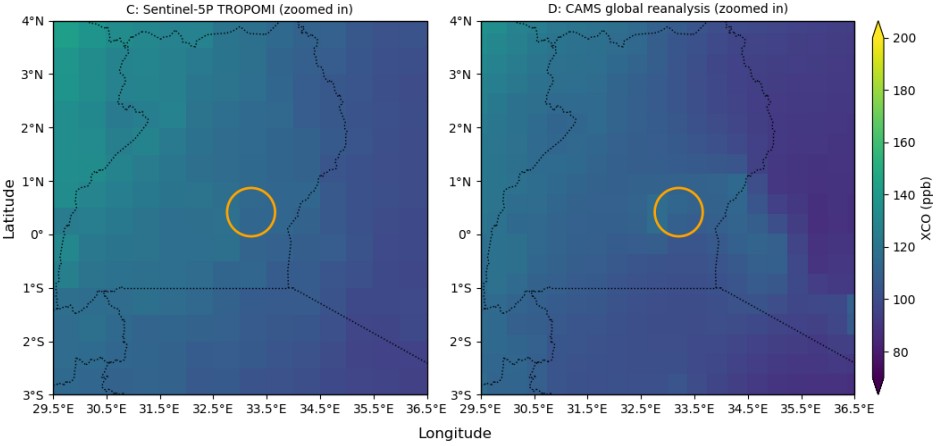

**Figure 6.** $X_{CO}$ over East Africa retrieved from Sentinel-5P TROPOMI observations (Panel A, see Section 3.2) and calculated from a global CAMS inversion (Panel B, see Section 3.3 for details), averaged over the EM27/SUN measurement period (23$^{rd}$ January to 19$^{th}$ April 2020.) The TROPOMI data are spatially binned into a 1x1 degree grid. The circles indicate the co-location criteria (50 km radius for $X_{CO}$, centred on the measurement site in Jinja) used in the comparisons described in Section 4.3. Panels C and D show the same data as panels A and B respectively, zoomed in on the co-location region.

EDGAR v4.3.2 database for anthropogenic emissions (Janssens-Maenhout et al., 2019), the WetCHARTs dataset for emissions from wetlands (Bloom et al., 2017), and the GFAS database for daily biomass burning emissions (Global Fire Assimilation System, Kaiser et al. (2012)). The boundary conditions for the nested domain come from a global GEOS-Chem model run at lower spatial resolution ($2.0° \times 2.5°$). An ensemble Kalman Filter system (Hunt et al., 2007) is then used to perform the




inversion, taking into account column $CH_4$ concentrations from TROPOMI (Lunt et al., 2021), which gives us estimates of the methane emissions within the nested domain along with the model-derived atmospheric concentrations. A subset of the mean $X_{CH_4}$ values for the measurement period calculated from the output of this high-resolution regional inversion, covering the region surrounding the Jinja site ($-3.0$ to $+3.0°$N, $+28.0$ to $+36.0°$E), are shown in Figure 5B.

The model dataset we use in this study for carbon monoxide is the Copernicus Atmosphere Monitoring Service (CAMS) global reanalysis dataset (Inness et al., 2019), which covers the period from January 2003 to December 2021 with a spatial resolution of approximately $80\,km$ (interpolated onto a regular $0.75° \times 0.75°$ grid) and 60 vertical levels. A 4D-VAR assimilation framework (Rabier et al., 2000; Hollingsworth et al., 2008) is used to produce the reanalysis, which is described in detail for carbon monoxide by Flemming et al. (2017). Total column carbon monoxide data retrieved from the MOPITT instrument
on board the NASA Terra satellite is used as input for the reanalysis (Deeter et al., 2014). Figure 6B shows the mean column concentration of carbon monoxide over East Africa for the measurement period, as calculated from the CAMS reanalysis output.

## 4    Comparisons of the EM27/SUN total column data with satellite and model data sets

In this section we show how the total column concentrations observed using the EM27/SUN in Jinja compare with both
satellite and model datasets, considering each species in turn. Before comparing the EM27/SUN and satellite data, we need to take into account that each retrieval algorithm used provides an estimate of the total column concentration that is based on different *a priori* information. Following the theory underpinning the optimal estimation retrieval method as described by Rodgers (2000), we correct for the different *a priori* profiles used in PROFFAST (see Section 2.3) and the respective satellite algorithms following Equation 3 in Dils et al. (2014), which assumes the ground-based *a priori* as the common *a priori* profile
when validating satellite GOSAT data and ground-based TCCON total column concentrations of carbon dioxide and methane:

$$x_{\mathrm{cor}} = x + \frac{1}{m_0} \sum_i m_i \left( A_i - 1 \right) \left( ap_{i,sat} - ap_{i,EM27} \right). \tag{3}$$

Here, $x_{\mathrm{cor}}$ and $x$ are the *a priori*-corrected and uncorrected dry-air total column concentrations; $i$ is the vertical layer index, with corresponding mass of dry air $m_i$ contained within the layer (derived from $\Delta p_i / g_i$, where $\Delta p_i$ is the dry air pressure difference over layer $i$ and $g_i$ is the acceleration due to gravity at that height). $m_0$ is the total dry air mass of the atmospheric column, obtained by taking the sum of $m_i$ over all layers. $A_i$ is the column averaging kernel used by the satellite retrieval
algorithm, and finally $ap_{i,sat}$ and $ap_{i,EM27}$ are the *a priori* dry air concentrations in layer $i$ assumed by the satellite and EM27/SUN retrieval algorithms respectively.

For these comparisons, we use co-location criteria which represent a trade-off between ensuring there are enough measurement days to be able to make meaningful conclusions from the observed EM27/SUN vs. satellite/model differences, whilst also
ensuring that we are spatially comparing like with like. In the case of carbon dioxide and methane retrievals from OCO-2/3 and Sentinel-5P TROPOMI respectively, we employ a wider co-location radius ($300\,km$) than used in the validation study of





Sha et al. (2021) for example, as the cloudy conditions commonly encountered in the tropics limit the number of successful satellite retrievals. The close proximity of Lake Victoria to the south of the measurement site also has an impact here, since the low albedo of the lake surface at shortwave infrared wavelengths reduces the intensity of the observed signal below the level

where a successful $X_{CO_2}$ or $X_{CH_4}$ retrieval is possible. We also split the time period in two, to check whether there is a notable difference in the comparisons as a result of the onset of the 'long rains' in March (see Section 2). In the following sections the two subsets are labelled 'dry', corresponding to January and February, and 'rainy', corresponding to the long rains period from March onwards.

To assess whether an observed difference between the EM27/SUN column concentrations and the satellite or model data are

statistically significant, we apply the t-test to the mean difference between the two datasets. This tests the null hypothesis that the expected value of the sample of differences is equal to zero (i.e. column concentrations observed by the EM27/SUN are equal to those observed by satellite or calculated by model). In the summary tables below we highlight in bold the instances where the p-value is less than 0.05, indicating a confidence level of $95\%$ or greater that the null hypothesis is false. The t-score produced by the test is the mean difference divided by the standard error.

### 4.1 Comparison of EM27/SUN $X_{CO_2}$ with OCO-2/-3 and GEOS-Chem

Here, we compare our EM27/SUN $X_{CO_2}$ values with those retrieved from OCO-2 and OCO-3 observations (*a priori*-corrected as described above), and obtained from a global GEOS-Chem $CO_2$ inversion which assimilates OCO-2 v10r data (see Section 3.3 for details). For the comparison we take OCO-2 and OCO-3 soundings (see Section 3.1), and GEOS-Chem grid points, within a $300\,km$ radius of the EM27/SUN location, and calculate the median $X_{CO_2}$ for each day. For $X_{CO_2}$ we use

all EM27/SUN data points, regardless of the time of day, in order to maximise the number of days of coincident OCO-2 and OCO-3 observations, and to take into account the varying OCO-3 overpass time. We also limit the OCO-2/3 vs. EM27/SUN comparison to days where there are at least five $X_{CO_2}$ soundings of sufficient quality that meet the co-location criteria described here. Figure 7 shows time series of these data, along with scatter plots directly comparing the EM27/SUN daily $X_{CO_2}$ with the satellite and model datasets. The statistics of the $X_{CO_2}$ comparisons are summarised in Table 1. The mean (standard

deviation) *a priori* profile corrections given by Equation 3 are $-0.238\,(0.013)\,ppm$ and $-0.373\,(0.082)\,ppm$ for OCO-2 and OCO-3 respectively.

Whilst acknowledging that there are only a few days during the measurement period where OCO-2 and OCO-3 data can be compared with our EM27/SUN measurements, the data that we have available suggests that during this period the $X_{CO_2}$ from OCO-2 is biased low with respect to that from the EM27/SUN, by $1.20\,ppm$ (standard deviation is $1.05\,ppm$), with a

confidence level of $98\%$. The $X_{CO_2}$ data from OCO-3 are also lower than what we observe from the EM27/SUN, however there are insufficient days of coincident observations during the measurement period to conclude that there is a statistically significant difference between the two.

The GEOS-Chem model columns are also generally biased low with respect to the EM27/SUN data, and for our measurement period the inversion is insensitive to whether both satellite and *in-situ* or only *in-situ* data are assimilated. We see in both

cases that these biases are primarily observed during the 'dry' period, where there are statistically significant differences from





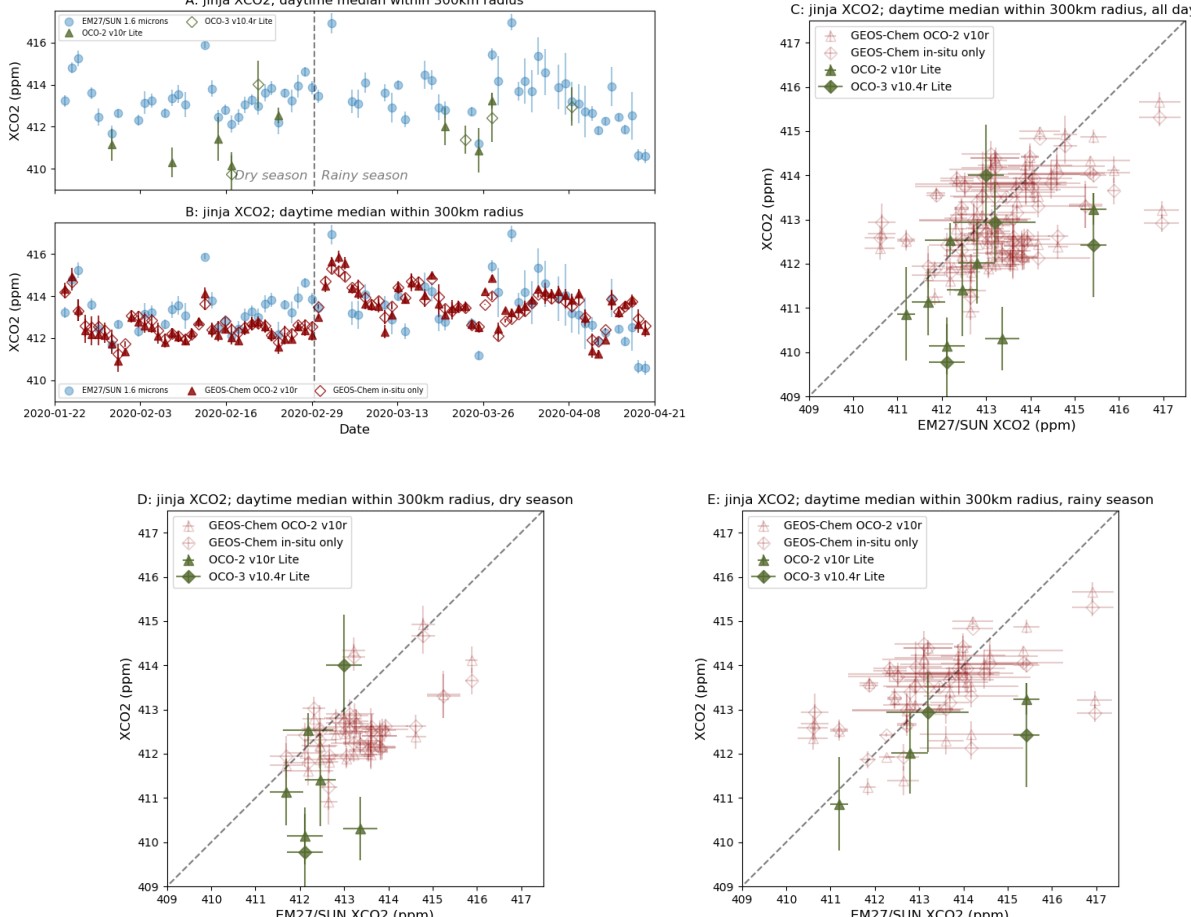

**Figure 7.** Panel A: daily median values of $X_{CO_2}$ retrieved from EM27/SUN (blue circles), OCO-2 (olive filled triangles) and OCO-3 (olive open diamonds) measurements, error bars show the inter-quartile range. Panel B: same as Panel A, but with global GEOS-Chem model data (red filled triangles are with OCO-2 data assimilated, red open diamonds use in-situ observations only). Panel C: scatter plot showing EM27/SUN daily median observations vs. OCO-2 (olive filled triangles), OCO-3 (olive filled diamonds) and global GEOS-Chem (red open triangles with OCO-2 assimilated, red open diamonds with in-situ data only), error bars show the inter-quartile range. We use a co-location radius of 300 km. Panel D: same as Panel C, but including data from the 'dry' season only (January and February 2020). Panel E: same as Panel C, but including data from the 'rainy' season only (March and April 2020).

the EM27/SUN columns of $-0.86\,\mathrm{ppm}$ and $-0.72\,\mathrm{ppm}$ respectively (standard deviations are $0.76\,\mathrm{ppm}$ and $0.79\,\mathrm{ppm}$). The differences are most clear during the last week of February 2020 (see Figure 7B), and suggest the possibility of local sources not captured by the relatively coarse ($2.0° \times 2.5°$ latitude-longitude grid) global configuration of GEOS-Chem used here.



**Table 1.** Mean and standard deviation of the differences ($\Delta X_{CO_2}$ in ppm) between daily median EM27/SUN and satellite/model $X_{CO_2}$. The 'dry' subset includes data from January and February 2020, whilst the 'rainy' subset covers data from March and April 2020.

| Satellite/model | Subset | Number of days | Total number of soundings | Mean $\Delta X_{CO_2}$ [ppm] | Std $\Delta X_{CO_2}$ [ppm] | t-score | p-value |
|---|---|---|---|---|---|---|---|
| **OCO-2 v10r** | **All** | **8** | **1725** | **-1.20** | **1.05** | **-3.03** | **0.019** |
| | Dry | 5 | 1173 | -1.26 | 1.17 | -2.15 | 0.098 |
| | Rainy | 3 | 552 | -1.10 | 0.79 | -1.97 | 0.19 |
| **OCO-3 v10.4r** | All | 4 | 324 | -1.15 | 1.61 | -1.23 | 0.31 |
| | Dry | 2 | 248 | -0.67 | 1.68 | -0.40 | 0.76 |
| | Rainy | 2 | 76 | -1.62 | 1.37 | -1.18 | 0.45 |
| **GEOS-Chem including OCO-2** | **All** | **68** | **n/a** | **-0.35** | **1.08** | **-2.65** | **0.0099** |
| | **Dry** | **31** | n/a | **-0.86** | **0.76** | **-6.21** | **$7.74 \times 10^{-7}$** |
| | Rainy | 37 | n/a | 0.077 | 1.13 | 0.41 | 0.68 |
| **GEOS-Chem in-situ only** | **All** | **68** | **n/a** | **-0.28** | **1.12** | **-2.03** | **0.046** |
| | **Dry** | **31** | n/a | **-0.72** | **0.79** | **-4.99** | **$2.42 \times 10^{-5}$** |
| | Rainy | 37 | n/a | 0.095 | 1.21 | 0.47 | 0.64 |

### 4.2 Comparison of EM27/SUN $X_{CH_4}$ with TROPOMI and GEOS-Chem

For $X_{CH_4}$ we compare the EM27/SUN column concentrations with *a priori*-corrected data from Sentinel-5P TROPOMI observations (see Section 3.2), and column concentrations calculated using *a priori* and *a posteriori* emissions from the high resolution GEOS-Chem inversion (see Section 3.3 for details of the model run). A $300\,km$ co-location radius is used for both satellite and model data, and we only use EM27/SUN data and GEOS-Chem time steps within $\pm 2\,hours$ of the Sentinel-5P overpass time, to calculate the median $X_{CH_4}$ for each day. We also restrict the TROPOMI vs. EM27/SUN comparison to days

where there are at least five $X_{CH_4}$ soundings of sufficient quality meeting the spatial and temporal co-location criteria described here. Figure 8 shows time series of these data, along with scatter plots directly comparing the EM27/SUN daily $X_{CH_4}$ with the satellite and model datasets. The statistics of the $X_{CH_4}$ comparisons are summarised in Table 2. The mean (standard deviation) *a priori* profile correction applied to the Sentinel-5P TROPOMI data, given by Equation 3, is $+1.56\,(0.15)\,ppb$.

The short measurement period limits the number of days where comparisons can be made between the ground-based and
satellite retrievals of $X_{CH_4}$. In the data we have, the TROPOMI retrievals are lower than the EM27/SUN columns by a mean of $8.33\,ppb$, albeit within the standard deviation $(10.5\,ppb)$ in the data. The p-value of 0.0741 indicates that despite the low number of measurement days we could use in the comparison, this difference still has some statistical significance, albeit not at the $95\%$ confidence level that we are using as a threshold for rejecting the null hypothesis here.





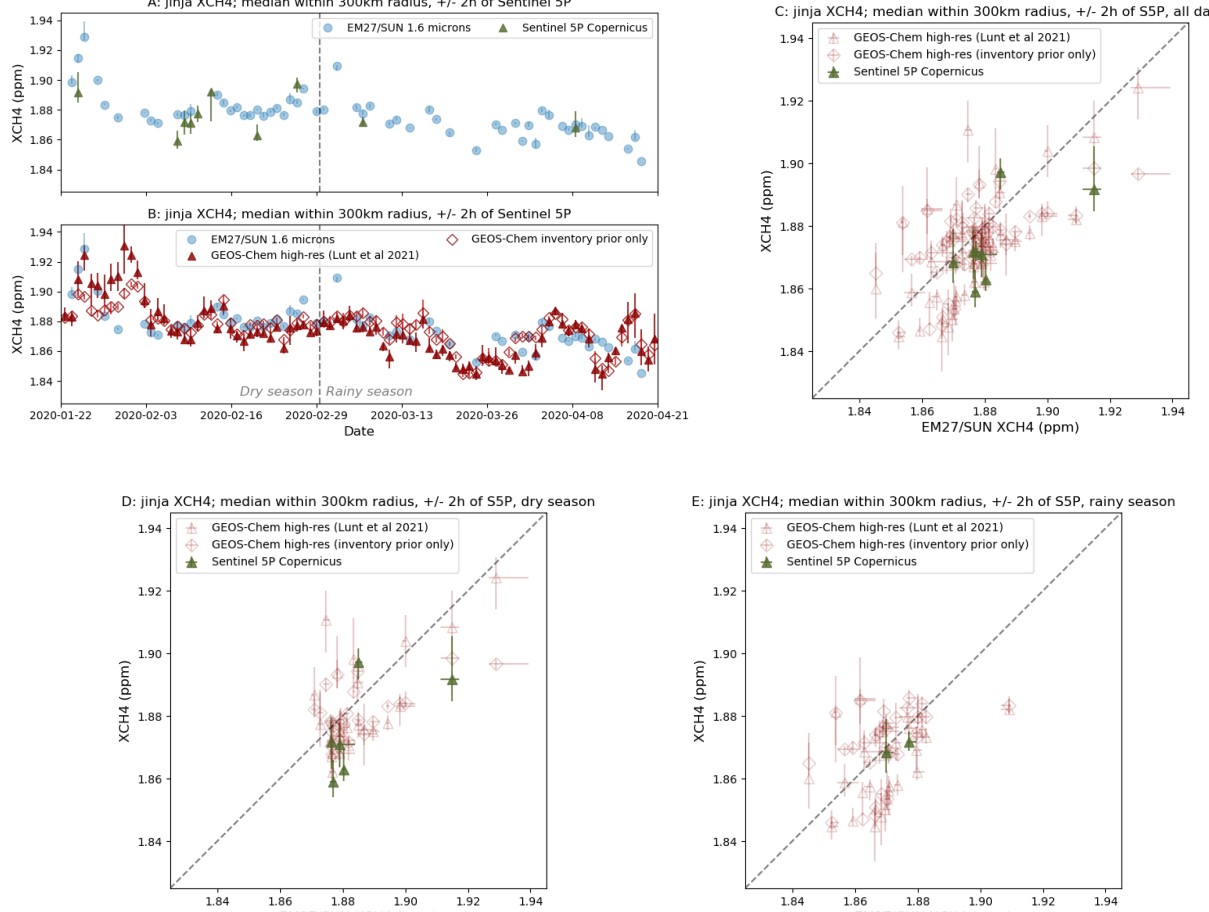

**Figure 8.** Panel A: daily median values of $X_{CH_4}$ retrieved from EM27/SUN (blue) and Sentinel-5P TROPOMI (dark green) measurements, bars show the inter-quartile range. Panel B: same as Panel A, but with high resolution regional GEOS-Chem model data (darker shade of red is with S5P data assimilated, lighter shade uses inventory emissions only). Panel C: scatter plot showing EM27/SUN observations vs. TROPOMI (dark green filled circles) and high resolution regional GEOS-Chem (red open circles with S5P assimilated, blue diamonds with inventory emissions only), error bars show the inter-quartile range. We use a co-location radius of $300\,km$, and only consider data and model output within $\pm 2\,hours$ of the Sentinel-5P overpass at 1030 UTC. Panel D: same as Panel C, but including data from the 'dry' period only (January and February 2020). Panel E: same as Panel C, but including data from the 'rainy' period only (March and April 2020).

The data from the GEOS-Chem high resolution inversions show better agreement with the EM27/SUN data in terms of the mean differences. The difference is slightly greater ($-3.80\,ppb$ compared with $-1.15\,ppb$) when the *a posteriori* emissions incorporating TROPOMI $X_{CH_4}$ are used, though the difference between the two is well within their respective standard deviations. The only comparison where there is a statistically significant difference from the EM27/SUN columns is that with the





**Table 2.** Mean and standard deviation of the differences ($\Delta X_{CH_4}$ in ppb) between daily median EM27/SUN and satellite/model $X_{CH_4}$. The 'dry' subset includes data from January and February 2020, whilst the 'rainy' subset covers data from March and April 2020.

| Satellite/model | Subset | Number of days | Total number of soundings | Mean $\Delta X_{CH_4}$ [ppb] | Std $\Delta X_{CH_4}$ [ppb] | t-score | p-value |
|---|---|---|---|---|---|---|---|
| **S5P Copernicus** | All | 8 | 790 | -8.33 | 10.5 | -2.10 | 0.074 |
| | Dry | 6 | 726 | -9.89 | 11.7 | -1.90 | 0.12 |
| | Rainy | 2 | 64 | -3.66 | 1.98 | -1.85 | 0.32 |
| **GEOSChem HR including S5P data** | **All** | **58** | **n/a** | **-3.80** | **12.5** | **-2.30** | **0.025** |
| | Dry | 27 | n/a | -2.85 | 11.9 | -1.28 | 0.21 |
| | Rainy | 31 | n/a | -4.53 | 13.0 | -1.91 | 0.066 |
| **GEOSChem HR inventory only** | All | 58 | n/a | -1.15 | 11.6 | -0.75 | 0.46 |
| | Dry | 27 | n/a | -3.59 | 10.7 | -1.70 | 0.10 |
| | Rainy | 31 | n/a | 0.90 | 12.2 | 0.41 | 0.69 |

GEOS-Chem simulation using *a posteriori* emissions, though there is not sufficient data to attribute this to either the 'dry' or the 'rainy' periods considered here.

There are a couple of possible explanations for differences seen between the EM27/SUN and GEOS-Chem $X_{CH_4}$ data. Firstly, the posterior scale factors which are applied to the prior emission fields have an exponential correlation length scale of 50 km, meaning that smaller scale variations in the emissions that influence the EM27/SUN measurements may not be reflected in the differences between the posterior and prior inversions. It is also worth noting that the *a posteriori* inversion minimises the residual to all TROPOMI $X_{CH_4}$ data within the larger inversion domain, rather than this specific grid box centred on Jinja. We can see from the EM27/SUN vs. TROPOMI comparison in Figure 8 that there are only a limited number of TROPOMI data available to constrain emissions during the measurement period, such that emissions local to the site are unlikely to be well represented in the inversion.

### 4.3 Comparison of EM27/SUN $X_{CO}$ with TROPOMI and CAMS

In the final part of this section we compare $X_{CO}$ retrieved from the EM27/SUN ground-based observations with $X_{CO}$ from Sentinel-5P TROPOMI data (see Section 3.2), and from the output of the global CAMS inversion (see Section 3.3 for details). The greater number of soundings with successful retrievals of $X_{CO}$ allows us to apply a tighter 50 km co-location radius to the satellite and model data. As for $X_{CH_4}$, we only use EM27/SUN data and GEOS-Chem time steps within $\pm 2$ hours of the Sentinel-5P overpass time, to calculate the median $X_{CO}$ value for each day. In addition we further restrict the TROPOMI





vs. EM27/SUN comparison to days where there are at least five $X_{CO}$ soundings of sufficient quality meeting these spatial

405 and temporal co-location criteria. Figure 9 shows time series of these data, along with scatter plots directly comparing the EM27/SUN daily $X_{CO}$ with the satellite and model datasets. The statistics of the $X_{CO}$ comparisons are summarised in Table 3. The mean (standard deviation) *a priori* profile correction applied to the Sentinel-5P TROPOMI data, given by Equation 3, is $-2.84\,(3.27)\,\mathrm{ppb}$.

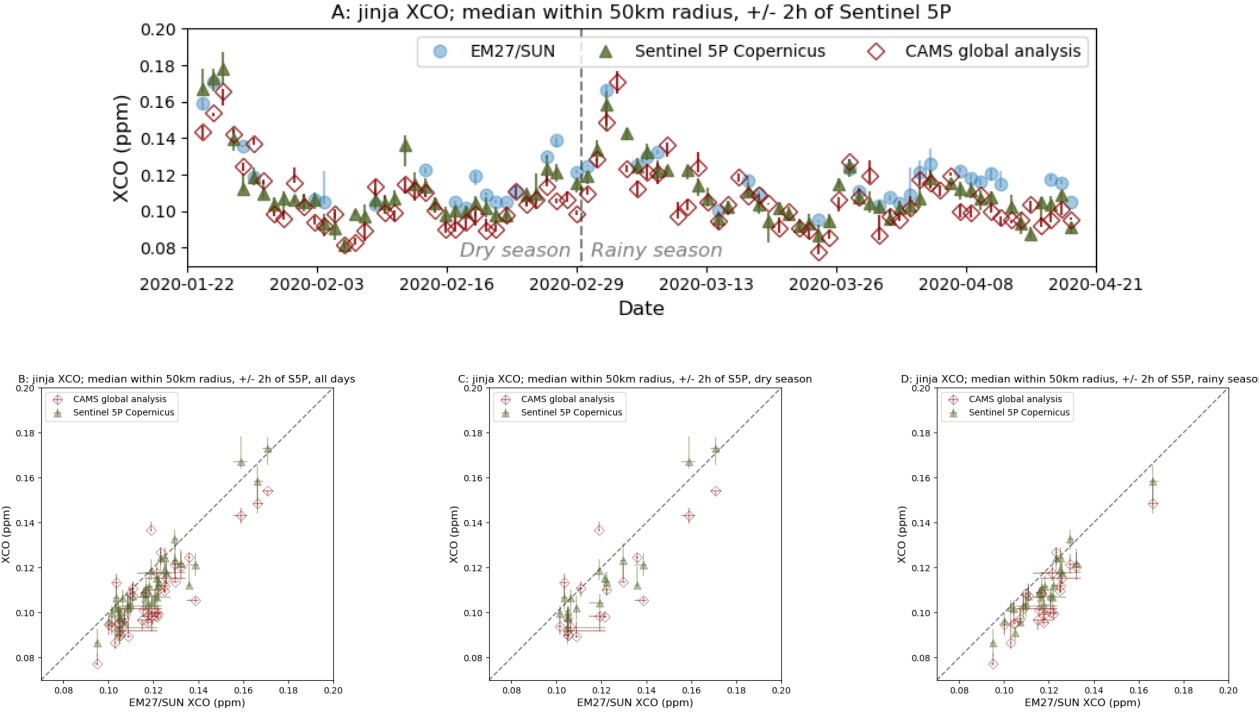

**Figure 9.** Panel A: daily median values of $X_{CO}$ retrieved from EM27/SUN (blue) and Sentinel-5P TROPOMI (dark green) measurements, and obtained from CAMS global analysis output (open red circles), bars show the inter-quartile range. Panel B: scatter plot showing EM27/SUN observations vs. TROPOMI (dark green filled circles) and global CAMS analysis data (red open circles), error bars show the inter-quartile range. We use a co-location radius of $50\,\mathrm{km}$, and only consider data and model output within 2 hours of the Sentinel-5P overpass at 1030 UTC. Panel C: same as Panel B, but including data from the 'dry' period only (January and February 2020). Panel D: same as Panel B, but including data from the 'rainy' period only (March and April 2020).

As discussed in Section 3.2, the availability of $X_{CO}$ data from TROPOMI in partially cloudy conditions means that we

410 have a greater number of measurement days suitable for comparison compared with $X_{CO_2}$ and $X_{CH_4}$. We find a statistically significant difference between our ground-based observations and the TROPOMI satellite data, with the $X_{CO}$ from TROPOMI being biased lower than that from the EM27/SUN by a mean value of $3.68\,\mathrm{ppb}$, which falls within the standard deviation $(7.00\,\mathrm{ppb})$ of the bias. There is also a difference when separating the 'dry' and 'rainy' periods, with the TROPOMI data in the 'rainy' period covering March and April 2020 showing a greater low bias with respect to the EM27/SUN columns than that

 

**Table 3.** Mean and standard deviation of the differences ($\Delta X_{CO}$ in ppb) between daily median EM27/SUN and satellite/model $X_{CO}$. The 'dry' subset includes data from January and February 2020, whilst the 'rainy' subset covers data from March and April 2020.

| Satellite/model | Subset | Number of days | Total number of soundings | Mean $\Delta X_{CO}$ [ppb] | Std $\Delta X_{CO}$ [ppb] | t-score | p-value |
|---|---|---|---|---|---|---|---|
| S5P Copernicus | All | 41 | 4738 | -3.68 | 7.00 | $-6.79$ | $3.63 \times 10^{-8}$ |
| | Dry | 17 | 2323 | -0.45 | 7.99 | $-3.19$ | 0.0057 |
| | Rainy | 24 | 2415 | -5.97 | 5.08 | $-7.01$ | $3.80 \times 10^{-7}$ |
| CAMS global analysis | All | 43 | n/a | -11.7 | 8.94 | $-8.48$ | $1.22 \times 10^{-10}$ |
| | Dry | 18 | n/a | -11.8 | 11.4 | $-4.29$ | $4.97 \times 10^{-4}$ |
| | Rainy | 25 | n/a | -11.6 | 6.65 | $-8.54$ | $9.69 \times 10^{-9}$ |

shown in the earlier 'dry' period. We note however that these differences are still well within the mission data requirements for the TROPOMI carbon monoxide data product, which stipulate that the bias should be less than 15% and the random error less than 10% (Landgraf et al., 2016).

The $X_{CO}$ values from the CAMS global model are also significantly low with respect to the ground-based EM27/SUN data, by a mean value of 11.7 ppb. This mean bias is greater than the standard deviation (8.94 ppb) both throughout our measurement period, and when separating into 'dry' and 'rainy' periods, suggesting that the CAMS global model may not be taking into account all local sources of carbon monoxide. In addition, recent work by Inness et al. (2022) has shown that assimilating TROPOMI carbon monoxide data into the CAMS system (in addition to the satellite data from MOPITT and IASI that is already assimilated in this version) increases the CAMS carbon monoxide columns by 8% on average, which would bring the CAMS model output into closer agreement with our EM27/SUN observations.

## 5 Conclusions and outlook

In this paper, we describe the first ground-based remote sensing observations of total column greenhouse gas concentrations to have been performed in the Tropical East Africa region. We set up a Bruker EM27/SUN spectrometer at the headquarters of the National Fisheries Resources Research Institute in Jinja, Uganda, in January 2020. An automated enclosure for the instrument, designed and built by the Environmental Sensing and Modelling Group at the Technical University of Munich, allowed us to operate the instrument remotely and autonomously for a period of three months, providing a temporal density of greenhouse gas column data over this period that would have been challenging to achieve manually. The combined performance of the instrument and enclosure shown in this paper demonstrates the possibility to deploy EM27/SUN instruments as validation sites for satellite greenhouse gas retrievals, in parts of the world where it would be logistically difficult to establish new sites to extend established ground-based validation networks such as TCCON.





The ground-based measurements of carbon dioxide, methane, and carbon monoxide column concentrations that we have acquired using the EM27/SUN and automated enclosure allow us, for the first time, to see how well satellite and model datasets have performed in observing or calculating these concentrations over Uganda during our measurement period. For carbon dioxide, we find statistically significant differences between the EM27/SUN and OCO-2 $X_{CO_2}$ (OCO-2 lower by a mean of $1.20\,\mathrm{ppm}$, with standard deviation $1.05\,\mathrm{ppm}$), and between the EM27/SUN and the global GEOS-Chem inversion we

use for this study – irrespective of whether OCO-2 data has been assimilated (GEOS-Chem lower by a mean of $0.35\,(1.08)\,\mathrm{ppm}$ with, and $0.28\,(1.12)\,\mathrm{ppm}$ without OCO-2 included in the inversion). We do not observe a statistically significant difference between EM27/SUN and OCO-3 $X_{CO_2}$ for this dataset (OCO-3 lower by a mean of $1.15\,(1.61)\,\mathrm{ppm}$). In the case of $X_{CH_4}$, we do not see a statistically significant difference between the S5P TROPOMI and EM27/SUN data (S5P lower by a mean of $8.33\,(10.5)\,\mathrm{ppb}$). We do however see a significant difference between the EM27/SUN and the high resolution GEOS-Chem

inversion we use in this study, which incorporates S5P TROPOMI data (GEOS-Chem lower by a mean of $3.80\,(12.5)\,\mathrm{ppb}$). This may be a result of the *a posteriori* inversion being set up to minimise the residual to all TROPOMI $X_{CH_4}$ data within the larger inversion domain, rather than just data within the specific grid box centred on Jinja where TROPOMI methane soundings are relatively scarce. This means that emissions local to the measurement site are unlikely to be well represented in the inversion.

In the case of carbon monoxide from S5P TROPOMI, the quality flagging of the column concentration retrievals is much less sensitive to cloud cover, such that there were many more days with coincident observations that we could compare with our EM27/SUN measurements. Even over a three month period, there was sufficient data to be able to conclude that the carbon monoxide columns from S5P TROPOMI were biased low with respect to the EM27/SUN data by a mean value of $3.68\,\mathrm{ppb}$ (standard deviation $7.00\,\mathrm{ppb}$). This is still well within the mission data requirements for the TROPOMI carbon monoxide

data product, which stipulate that the bias should be less than $15\%$ and the random error less than $10\%$ (Landgraf et al., 2016). We also see a statistically significant difference between our EM27/SUN measurements and the CAMS global analysis (CAMS $X_{CO}$ lower by a mean of $11.7\,\mathrm{ppb}$, with standard deviation $8.94\,\mathrm{ppb}$), suggesting that the CAMS global model may not be taking into account all local sources of carbon monoxide. Recent work by Inness et al. (2022) has, however, shown that assimilating TROPOMI carbon monoxide data into the CAMS system (in addition to the satellite data from MOPITT

and IASI that is already assimilated in this version) would increase the CAMS carbon monoxide columns by $8\%$ on average, which if applicable to tropical East Africa would bring the CAMS model output into closer agreement with our EM27/SUN observations.

  An important aspect of this work is the comparison with atmospheric chemistry and transport model output. Models and reanalyses such as GEOS-Chem and CAMS provide a means of studying atmospheric processes where observations are not

available. This is of particular relevance in tropical Africa (e.g. Lunt et al. (2019, 2021); Palmer et al. (2019); Feng et al. (2022)), where ground-based observations of greenhouse gases are scarce and the data coverage provided by satellites is often limited by cloud cover. Ground-based column concentration observations such as those presented in this study provide data that can be used to evaluate these models which, unlike in-situ measurements, are not overly sensitive to emission sources on a local scale. Our results show that only three months of measurements can be sufficient to demonstrate the effectiveness of





these models at this time of year, whilst also highlighting short periods where there are discrepancies to be investigated further. A comprehensive validation of the models would require at least a whole calendar year of observations. Figure 10 shows radial histograms of the wind direction for each month of the year 2020. These demonstrate how the typical wind direction at $800\,\mathrm{hPa}$, in the lower troposphere where the retrieved column concentrations are generally most sensitive, varies throughout the year. During the measurement period for this study, as summarised by the radial histogram in Figure 10A, the wind has

most frequently blown from the north (particularly in January and February) and from the south east (from late February to April). A full year of measurements would be more representative of the variety of atmospheric conditions we would expect to observe from satellites or estimate from models, and would give us greater confidence in the performances of the retrieval algorithms and model calculations respectively. The period from October to December would be particularly interesting to focus on in the future, as the typically northerly winds we see at that time of year (Figure 10B) coincide with the 'short rains',

the intensity of which Lunt et al. (2021) have linked to changes in methane emissions from the Sudd wetlands, located in South Sudan to the north of the measurement site.

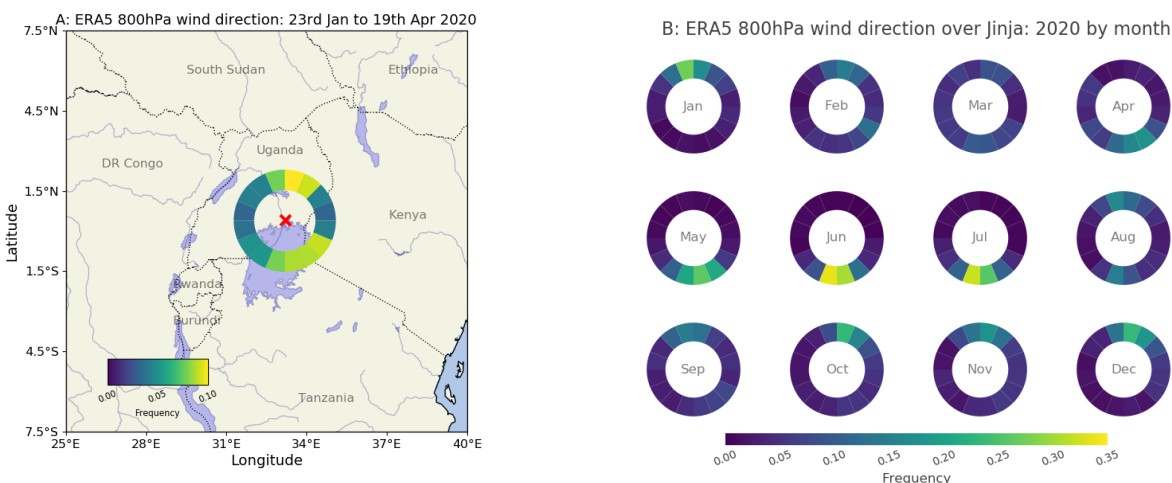

**Figure 10.** Panel A: radial histogram showing frequency of wind direction in the ERA5 reanalysis at $800\,\mathrm{hPa}$ above Jinja during the measurement period for this study. Panel B: radial histograms showing the frequency of wind direction at $800\,\mathrm{hPa}$ in ERA5 over Jinja for each month in 2020. The angles of the histogram segments correspond to the direction that the wind is coming from. Note that we use different colour scales in Panels A and B.

In summary, this study demonstrates the feasibility of a longer-term, autonomous deployment of the EM27/SUN instrument in a tropical environment, through the use of an automated weatherproof enclosure. This EM27/SUN plus enclosure system allows us to meet the goal of seasonal observations in support of studies focusing on the tropical carbon cycle, and the validation

of greenhouse gas column concentration data from satellite retrievals and model calculations in the tropical East Africa region.



*Code and data availability.* The EM27/SUN column data and GEOS-Chem data will be made available on the Centre for Environmental Data Analysis archive. The latest version of the Pyra software used to control the automated weatherproof enclosure is available on GitHub (https://github.com/tum-esm/pyra). The latest version of the PROFFAST interferogram processing and analysis code is available from https://www.imk-asf.kit.edu/english/3225.php. The L2 column carbon dioxide data from OCO-2 and OCO-3 are available from the

Goddard Earth Sciences Data and Information Services Centre (https://disc.gsfc.nasa.gov/datasets). The Sentinel-5P TROPOMI column methane and carbon monoxide data are available from the Sentinel-5P Pre-Operations Data Hub (https://s5phub.copernicus.eu/). The CAMS global reanalysis carbon monoxide concentration data are available from the CAMS Atmosphere Data Store (see https://www.ecmwf.int/en/research/climate-reanalysis/cams-reanalysis). The ERA5 reanalysis wind data are available from the Copernicus Climate Data Store (https://cds.climate.copernicus.eu/).

*Author contributions.* NH set up the measurements, remotely monitored the instrument performance during the study period, performed the EM27/SUN data processing and analysis, prepared the figures, and wrote the manuscript. HB conceived the study, advised on interpretation of the results, and reviewed the manuscript. WO provided and prepared the measurement site, helped set up the measurements, and oversaw the site once measurements were up and running. JC and FD developed and built the automated weatherproof enclosure for the EM27/SUN used in this study. MFL performed the GEOS-Chem methane model runs, advised on interpretation of the results, and reviewed

the manuscript. LF performed the GEOS-Chem carbon dioxide model runs and advised on their interpretation. PIP advised on the wider context and interpretation of the results, and reviewed the manuscript. FH devised the EM27/SUN instrument concept, and developed the PROFFAST retrieval code used in this study.

*Competing interests.* FH is a member of the editorial board of Atmospheric Measurement Techniques. The authors have no other competing interests to declare.

*Acknowledgements.* The authors would like to acknowledge the staff at the National Fisheries Resources Research Institute in Jinja, Uganda, for their support of the measurements that underpin this work. NH, HB, MFL, LF, and PIP acknowledge the funding provided by the Natural Environment Research Council for this work, through award ref. NE/N015681/1 'The Global Methane Budget'. NH, HB, LF, and PIP also acknowledge funding from the Natural Environment Research Council through the National Centre for Earth Observation (grant nos. NE/R016518/1 and NE/R000115/1). JC and FD acknowledge funding by the Deutsche Forschungsgemeinschaft (DGF, German

Research Foundation; grant nos. CH 1792/2-1 and INST 95/1544). FH acknowledges continuing support by the European Space Agency for development and operation of the COCCON central facility, specifically the COCCON-PROCEEDS projects (contract 4000121212/17/I-EF) and COCCON-OPERA (contract 4000140431/23/I-DT-Ir). This research used the ALICE High Performance Computing Facility at the University of Leicester, along with JASMIN (http://jasmin.ac.uk), the UK collaborative data analysis facility. The OCO-2 and OCO-3 data were produced by the OCO-2 project at the Jet Propulsion Laboratory, California Institute of Technology, and obtained from the OCO-2 data

archive maintained at the NASA Goddard Earth Science Data and Information Services Center.



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
