# Peer review of "Greenhouse gas column observations from a portable spectrometer in Uganda"

_Atmospheric Measurement Techniques, 2023_

## Author Response (AR1)

**AMT-2023-234 AUTHOR RESPONSES TO REFEREE COMMENTS**

**Referee Comment 1**

We would firstly like to thank the Anonymous Referee for their constructive feedback on the manuscript. As a result of their comments we have made the following changes to the final version of the manuscript:

On the choice of t-tests to assess statistical significance given the small sample size, there is literature concluding that "there are no principal objections to using a t-test with Ns as small as 2" (de Winter, 2019), although they do qualify this with "as long as the effect size is expected to be large". In these comparisons the relative differences between the EM27/SUN measurements and the satellite/model/reanalysis datasets are small, so we think that especially where we only have a handful of samples we are overstating the statistical significance of some of these differences. We agree with the referee that it would be better and clearer to present the comparisons in terms of mean difference, standard deviation and relative difference -- we have updated the tables, and the results section to reflect this. The second comment regarding the justifications for not having statistically significant differences is related to this, so we have revisited and updated these parts of the results sections to avoid overinterpreting statistical significance (or lack thereof).

We have also addressed the following typo/editing comments/suggestions:

1. Page 7 lines 186-187: I would replace the occurrences of "radiance spectra" with "solar absorption spectra." – **we have replaced these as suggested.**

2. Subsections 3.1 and 3.2: It could be clearer for the reader if the subsection titles included the species retrieved by these satellites. For example, 3.1 could become "Orbiting Carbon Observatory (OCO-2 and OCO-3) XCO2 retrievals," and 3.2 "Sentinel-5 Precursor XCH4 and XCO retrievals." This is only a suggestion -- **we agree with this suggestion, and have renamed the subsections to make them clearer for the reader.**

3. CAMS: CAMS should be referred to as a (atmospheric composition) reanalysis as it is not strictly a model or an inversion. This should be harmonized through the manuscript -- **we have updated the manuscript to consistently refer to CAMS as a reanalysis throughout, as suggested.**

4. TROPOMI: I would replace all occurrences of "Sentinel-5P TROPOMI" (and similar occurrences, e.g., "S5P Copernicus" in Table 2) with "TROPOMI." Whatever is the author's choice, it should be harmonized throughout the manuscript -- **other than when it is first mentioned, where we make it clear the TROPOMI is on board Sentinel-5P, we have updated the**

**manuscript to consistently refer to TROPOMI throughout (including the Tables and in legends in Figures).**

5. Page 12, line 285, aren't -> are not -- **we have replaced as suggested.**

6. Tables 1, 2, and 3: The total number of soundings could be replaced by "Total number of satellite soundings" or "Total number of satellite retrievals." -- **in Tables 1, 2, and 3, we have replaced this column header with "Total number of satellite retrievals"**

Reference:

de Winter, J.C.F. (2019) "Using the Student's t-test with extremely small sample sizes," Practical Assessment, Research, and Evaluation: Vol. 18, Article 10. DOI: https://doi.org/10.7275/e4r6-dj05

**Referee Comment 2**

We would firstly like to thank Shima Bahramvash Shams for their constructive feedback on the manuscript. As a result of their comments we have made the following changes to the final version of the manuscript:

1. The referee suggests that there is insufficient detail provided about the instrument characteristics and retrieval. Whilst we feel this material is well covered in the references provided, we have added an appendix covering some of the characteristics mentioned (example spectra, information on spectral range, and plots of averaging kernels to illustrate vertical sensitivity) to help a reader unfamiliar with the instrument to understand the measurements.

2. We accept that there is a spatial smoothing effect that comes with co-location averaging. It is a common problem when comparing ground-based observations with satellite data that the satellite sampling pattern often does not fall exactly on the ground-based spectrometer location, which is why previous studies of a similar nature have taken the spatial co-location averaging approach (Inoue et al 2016, Wunch et al 2017, Sha et al 2021). To demonstrate the validity of this approach and justify the selection of the co-location radii used, we have added an appendix showing the impact of varying the coincidence criteria on the mean and standard deviation of the satellite vs. ground-based column concentration difference.

We have also addressed the minor comments, as follows:

-P4, L116: please add some references for the validation works using TCCON site -- **we have added appropriate references here (Inoue et al 2016, Wunch et al 2017, Sha et al 2021)**

-P8, L189: Please be specific about the sources of A priori for the retrieval, which model or climatology they are made off -- **we have added further detail and references on how the a priori profiles are generated here, as suggested.**
-Figure 3 and Table 1: I would change the left-hand side label from number of soundings to number of scan or successful retrieval as sounding is mostly referred to in situ measurement and it is confusing if there are in situ observation available -- **we have changed both of these to 'Successful Retrievals' to make the Figure and Table clearer to the reader, as suggested.**

References:

Inoue, M., Morino, I., Uchino, O., Nakatsuru, T., Yoshida, Y., Yokota, T., Wunch, D., Wennberg, P. O., Roehl, C. M., Griffith, D. W. T., Velazco, V. A., Deutscher, N. M., Warneke, T., Notholt, J., Robinson, J., Sherlock, V., Hase, F., Blumenstock, T., Rettinger, M., Sussmann, R., Kyrö, E., Kivi, R., Shiomi, K., Kawakami, S., De Mazière, M., Arnold, S. G., Feist, D. G., Barrow, E. A., Barney, J., Dubey, M., Schneider, M., Iraci, L. T., Podolske, J. R., Hillyard, P. W., Machida, T., Sawa, Y., Tsuboi, K., Matsueda, H., Sweeney, C., Tans, P. P., Andrews, A. E., Biraud, S. C., Fukuyama, Y., Pittman, J. V., Kort, E. A., and Tanaka, T.: Bias corrections of GOSAT SWIR $XCO_2$ and $XCH_4$ with TCCON data and their evaluation using aircraft measurement data, Atmos. Meas. Tech., 9, 3491–3512, https://doi.org/10.5194/amt-9-3491-2016, 2016.

Wunch, D., Wennberg, P. O., Osterman, G., Fisher, B., Naylor, B., Roehl, C. M., O'Dell, C., Mandrake, L., Viatte, C., Kiel, M., Griffith, D. W. T., Deutscher, N. M., Velazco, V. A., Notholt, J., Warneke, T., Petri, C., De Maziere, M., Sha, M. K., Sussmann, R., Rettinger, M., Pollard, D., Robinson, J., Morino, I., Uchino, O., Hase, F., Blumenstock, T., Feist, D. G., Arnold, S. G., Strong, K., Mendonca, J., Kivi, R., Heikkinen, P., Iraci, L., Podolske, J., Hillyard, P. W., Kawakami, S., Dubey, M. K., Parker, H. A., Sepulveda, E., García, O. E., Te, Y., Jeseck, P., Gunson, M. R., Crisp, D., and Eldering, A.: Comparisons of the Orbiting Carbon Observatory-2 (OCO-2) XCO2 measurements with TCCON, Atmos. Meas. Tech., 10, 2209–2238, https://doi.org/10.5194/amt-10-2209-2017, 2017.

Sha, M. K., Langerock, B., Blavier, J.-F. L., Blumenstock, T., Borsdorff, T., Buschmann, M., Dehn, A., De Mazière, M., Deutscher, N. M., Feist, D. G., García, O. E., Griffith, D. W. T., Grutter, M., Hannigan, J. W., Hase, F., Heikkinen, P., Hermans, C., Iraci, L. T., Jeseck, P., Jones, N., Kivi, R., Kumps, N., Landgraf, J., Lorente, A., Mahieu, E., Makarova, M. V., Mellqvist, J., Metzger, J.-M., Morino, I., Nagahama, T., Notholt, J., Ohyama, H., Ortega, I., Palm, M., Petri, C., Pollard, D. F., Rettinger, M., Robinson, J., Roche, S., Roehl, C. M., Röhling, A. N., Rousogenous, C., Schneider, M., Shiomi, K., Smale, D., Stremme, W., Strong, K., Sussmann, R., Té, Y., Uchino, O., Velazco, V. A., Vigouroux, C., Vrekoussis, M., Wang, P., Warneke, T., Wizenberg, T., Wunch, D., Yamanouchi, S., Yang, Y., and Zhou, M.: Validation of methane and carbon monoxide from Sentinel-5 Precursor using TCCON and NDACC-IRWG stations, Atmos. Meas. Tech., 14, 6249–6304, https://doi.org/10.5194/amt-14-6249-2021, 2021.